# Heme Oxygenase 1: A Defensive Mediator in Kidney Diseases

**DOI:** 10.3390/ijms22042009

**Published:** 2021-02-18

**Authors:** Anne Grunenwald, Lubka T. Roumenina, Marie Frimat

**Affiliations:** 1Centre de Recherche des Cordeliers, INSERM, Sorbonne Université, Université de Paris, F-75006 Paris, France; anne.grunenwald@inserm.fr (A.G.); lubka.roumenina@sorbonne-universite.fr (L.T.R.); 2U1167-RID-AGE, Institut Pasteur de Lille, Inserm, Univ. Lille, F-59000 Lille, France; 3Nephrology Department, CHU Lille, Univ. Lille, F-59000 Lille, France

**Keywords:** heme-oxygenase-1, heme, kidney, hemolysis, rhabdomyolysis, toxicity, ischemia reperfusion

## Abstract

The incidence of kidney disease is rising, constituting a significant burden on the healthcare system and making identification of new therapeutic targets increasingly urgent. The heme oxygenase (HO) system performs an important function in the regulation of oxidative stress and inflammation and, via these mechanisms, is thought to play a role in the prevention of non-specific injuries following acute renal failure or resulting from chronic kidney disease. The expression of HO-1 is strongly inducible by a wide range of stimuli in the kidney, consequent to the kidney’s filtration role which means HO-1 is exposed to a wide range of endogenous and exogenous molecules, and it has been shown to be protective in a variety of nephropathological animal models. Interestingly, the positive effect of HO-1 occurs in both hemolysis- and rhabdomyolysis-dominated diseases, where the kidney is extensively exposed to heme (a major HO-1 inducer), as well as in non-heme-dependent diseases such as hypertension, diabetic nephropathy or progression to end-stage renal disease. This highlights the complexity of HO-1’s functions, which is also illustrated by the fact that, despite the abundance of preclinical data, no drug targeting HO-1 has so far been translated into clinical use. The objective of this review is to assess current knowledge relating HO-1’s role in the kidney and its potential interest as a nephroprotection agent. The potential therapeutic openings will be presented, in particular through the identification of clinical trials targeting this enzyme or its products.

## 1. Introduction

The number of people worldwide with chronic kidney disease (CKD), acute kidney injury (AKI) or requiring renal replacement therapy exceeds 850 million [1]. The forecasted incidence is a cause for concern, with CKD projected to become the world’s fifth leading cause of death by 2040 [2]. It is generally accepted that kidney diseases are a significant public health issue and that the need for a multimodal approach to curb this presumed evolution is urgently needed [3]. Of key importance is the identification of novel therapeutic targets, both to limit the impact of acute renal events and to ameliorate poor outcomes in CKD, across the range of underlying etiologies. Several candidates for nephroprotection are currently being studied [4,5]. The heme oxygenase (HO) system is one of the targets that could be used to protect kidney structures from damage caused by oxidative stress, limiting inflammation and the consequent accelerated ageing. 

HO was discovered in 1968 by Tenhunen and colleagues, who first described its ability to catalyze the breakdown of heme (Fe-protoporphyrin IX) to free iron, carbon monoxide (CO) and biliverdine, which is rapidly converted to bilirubin [6]. Two main isoforms of HO were described: inducible HO-1 and constitutive HO-2. Initially merely considered a recycling system for heme from aged red blood cells, HO has since been attributed a number of cytoprotective properties via its metabolites and downstream signaling, expanding its definition from that of *a “molecular wrecking ball” to a “mesmerizing” trigger of cellular events*, to quote Maines et al. [7].

Driven by its potential antioxidant and anti-inflammatory capacities, HO-1 is the subject of growing interest for its role in different human pathologies. Among the studied organs, the kidney has often been highlighted over the last 30 years, notably through the identification of a role for HO-1 in various renal, metabolic and vascular diseases of both an acute and chronic nature [8,9,10,11]. HO-1 is essential for efficient kidney function, as illustrated by kidney damage in humans with HO-1 deficiency and by numerous animal studies (described below). The kidney’s physiological functions specifically influence HO-1 regulation. Indeed, renal tissue is very susceptible to hypoxia [12], a major driver of HO-1 expression, and is particularly exposed to toxic molecules because of its filtration and reabsorption functions. This particularity of being a “detoxification” organ, shared with the liver, means the kidney is highly exposed to damage-associated molecular patterns (DAMPs) and stress mediators. Interestingly, the kidney possesses a capacity for the synthesis of albumin, alphaprotein and haptoglobin under ischemic or toxic ischemic damage, underlining the parallel with the liver [13,14]. Renal cells therefore require adaptation strategies to preserve their functions and integrity in such a harsh environment. The kidney is a major site of exposure to heme in the event of extravascular hemolysis or rhabdomyolysis, and hence also for the induction of HO-1 expression. Under these conditions, HO-1 overexpression is easily explained by the need to catabolize the free heme overload. However, HO-1’s involvement in kidney diseases goes beyond the framework of heme-related diseases, illustrating the complexity of its mechanisms of action which extend beyond heme catabolism alone [15]. 

The aim of this review is to summarize the large amount of experimental and clinical data describing the importance of HO-1 in the kidney, without forgetting that HO-2 is also significant: constitutively expressed in the kidney under homeostatic conditions, it is in the first line of defense against ischemia and other insults, and its renal cytoprotective effects have been demonstrated in heme protein- and ischemic-induced AKI [16]. Compared to the literature about HO-1, however, there are very few studies examining the functional significance of HO-2 in kidney diseases; therefore, this review will focus upon HO-1. After a general introduction on the characteristics of HO-1 in the kidney, we will describe the role of this enzyme in different renal cell populations and in kidney diseases. Given its primary function of recycling heme, we will approach these pathological states according to whether or not there is a massive release of free heme. Finally, current clinical applications will be presented, revealing the disparity between the profusion of available experimental data and the relative paucity of therapeutic applications.

## 2. About Heme Oxygenase-1 in Kidney

### 2.1. Why Is HO-1 Particularly Important for the Kidney?

Over the past few decades, the key role of the HO-1 system in the pathophysiology of kidney diseases has been supported by several studies. In the two HO-1-deficient patients reported up to date, the kidney was among the damaged organs. Both patients had hematuria and proteinuria. Kidney biopsies revealed increased mesangial proliferation and focal thickening of the capillary loops by light microscopy, and endothelial detachment in the glomerular capillary by electron microscopy [17,18]. Renal injuries were also present in both animal models invalidated for HMOX1, with some differences apparent between the mouse [19] and rat models [20]. In Hmox1^−/−^ mice, lack of functional heme oxygenase was responsible for enhanced susceptibility to oxidative stress [19], decreased hemophagocytosis of senescent red blood cells by tissue macrophages, increased hemolysis and redistribution of iron from splenic and hepatic macrophages to hepatocytes and renal proximal tubule cells [21]. Similarly, Hmox1^−/−^ rats exhibited anemia, splenomegaly, and prominent interstitial inflammatory cell infiltrates and fibrosis scarring in their kidneys. While glomerular lesions were anecdotal in the mouse model, Atsaves et al. found an increase in mesangial matrix and focal and segmental glomerulosclerotic lesions in Hmox1^−/−^ rats. These morphological findings were associated with increased blood urea nitrogen, serum creatinine and albuminuria, but interestingly there was no increase in iron deposits in the glomeruli, tubules or interstitium [20].

Spontaneous renal damage in both human and animal models invalidated for HMOX suggests that HO-1 is important for efficient renal function. The renal protective role of this enzyme has been confirmed in multiple models of renal challenge in HO-1 invalidation/inhibition or overexpression conditions [22]. However, the histological differences cited above illustrate the difficulties of transposing results from one species to another. Differences in the regulation of HO-1 between humans and mice are well described, motivating the generation of human hHO-1 BAC transgenic mice [23]. In this model, HO-1 is overexpressed, and it remains unknown as to whether this higher background expression may obscure subtle damage over longer periods of time, or even have adverse effects itself. 

A better understanding of how HO-1 functions in the kidney, and how its expression or function is controlled, is essential for our comprehension of kidney physiology and pathology but it is (as always) important to remember the limits of any study in its interpretation and translation: we will attempt to sustain this critical viewpoint throughout this review. 

### 2.2. Regulation of HO-1 Expression in Kidneys

HO-1 protein is ubiquitous, being anchored in the membrane of the endoplasmic reticulum and also localized in mitochondrial nuclei and caveoli, while the HO-1 gene (HMOX1), which encodes HO-1, is located on Chromosome 22. Many stimuli can modulate the transcription of HMOX1, reflecting the broad spectrum of DNA-binding motifs within its promoter and the large number of signaling pathways which lead to its transcription. The main regulator of the HMOX1 gene is nuclear erythroid factor 2 (Nrf2), whose activation depends on its interaction with the Kelch-type, ECH-associated protein (Keap1). The Keap–Nrf2 interaction promotes the degradation of Nrf2 (healthy state), while destabilization of this complex releases Nrf2 which translocates into the nucleus and upregulates certain genes, including HO-1 (stressed state). There are many other activators or repressors of HO-1 forming a complex biomolecular network (detailed in previous reviews [7,10]). Details on these are beyond the scope of this review and we will here focus upon the renal-specific features of HO-1 expression. 

In the kidney, HO-1 protein levels are undetectable under homeostatic conditions, except in the tubules where it nevertheless remains low [24]. Many stress conditions are known to up-regulate the renal transcription HO-1, including oxidative stress, heat shock, hypoxia, heavy metals and toxins (reviewed in Bolisetty et al. [8]). Under these conditions, HO-1 expression remains heterogenous among the different renal compartments. The kidney consists of two main areas: the output cortex and the medulla in the innermost region. The medulla contains most of the length of the nephrons, the functional components of the kidney that filter fluid from the blood. The renal cortex includes the glomeruli, those “tufts” of capillaries where plasma is filtered through the glomerular basement membrane. The filtered fluid then flows along the convoluted proximal tubule to the loop of Henle, and then to the convoluted distal tubule and collecting ducts, which drain into the ureter (Figure 1). Glomerular filtration and reabsorption of solutes (the kidney’s two major functions) are supported by renal blood flow, which is among the highest in the body, with kidneys receiving about 20 to 25% of cardiac output. Compared to the cortex, blood flow to the renal medulla is relatively low. This disparity is required to enable an effective urine concentrating mechanism and also explains the particular susceptibility of the medulla to decreased blood flow. Under stress conditions, HO-1 is thus more highly expressed in the medulla than the cortex [25]. Its expression is very strong in tubules (especially proximal tubules), but minimal or absent in glomeruli, as illustrated in Figure 2 in both hemolytic patients and heme-injected mice. However, it has been shown that prior induction of HO-1 within the glomeruli prevented subsequent development of nephrotoxic glomerulonephritis in a Lewis rat model, but the major producer of HO-1 in the glomeruli was the infiltrating macrophages not the intrinsic glomerular cells [26]. This lower capacity of glomeruli to express HO-1, which is a key cytoprotective mechanism, could explain glomeruli’s particular sensitivity to stressors in certain pathologies [27].

Taken together, these data show that HO-1 levels are normally undetectable in the kidney but that HO-1 is highly inducible in tubules during pathologies, notably via its main transcription factor Nrf2. The capacity of glomeruli to induce HO-1 is lower, however. 

### 2.3. HO-1 and Nephroprotection 

Like HO-2, HO-1 drives the NADPH-dependent addition of an oxygen molecule to the porphyrin ring of heme, thus catalyzing the oxidation of heme and the equimolar release of biliverdin, free iron, and carbon oxide (CO). All these events are cytoprotective by removing free heme, regulating iron efflux, and increasing bilirubin and CO levels. Indeed, the noxious effects of free heme are well described, particularly at the vascular site via direct toxicity, but also through the stimulation of pro-inflammatory signaling pathways (such as DAMPs) and the activation of the complement system [28]. Similar to endothelial cells, such adverse effects are reported with other cell types when massively exposed to heme. This is the case for human skeletal muscle fibers, in which exposure to heme induces contractile dysfunction [29], and renal tubular cells [30]. In addition, iron is notably toxic even at low concentrations. It participates in the Fenton reaction, producing hydroxyl, and can result in cell death by ferroptosis, a cell-death pathway characterized by an iron-dependent decrease in glutathione (GSH) levels and the accumulation of lipid hydroperoxides to lethal levels [31]. Thus, it has been reported that the cytoprotection of HO-1 was due, at least in part, to an increased iron flux [32], which following its release is either stored in ferritin or exported from macrophages to plasma for re-use via the transmembrane ferrous exporter ferroportin. CO counteracts vasoconstriction and has powerful anti-oxidant properties, as does biliverdin, which is enzymatically converted to bilirubin via biliverdin reductase [33].

Transposing these properties to the renal level, it is easy to understand research interest in harnessing this molecule in order to treat kidney disease. The positive effects have been described for almost 30 years, following the work of Nath et al., who reported that HO-1 conferred protection in a rat model of glycerol-induced, acute renal failure. In their model, a single prior injection of hemoglobin rapidly induced HO-1 messenger RNA and protein within the kidney, which prevented progression to renal failure [34]. The nephroprotective effects of HO-1 and its metabolites have since been reported in different models of renal disease [8,35]. The potential importance of HO-1 in the pathogenesis of kidney diseases is also suggested by the link between polymorphisms in the HO-1 gene and renal outcome. Individuals with shorter (GT)n repeats in the HO-1 promoter region have a higher transcriptional activity and thus higher HO-1 levels than individuals with longer (GT)n repeats [36]. Interestingly, long (GT)n repeats have been associated with worse prognosis in several contexts, namely, a higher risk for AKI after cardiac surgery or decreased renal function after kidney transplantation [37,38]. However, there is no evidence of a protective effect of short repeat (GT)n for graft or recipient survival following renal transplant [39], underlining the complexity of the mechanisms involved. The levels of HO-1 expression may also vary with age. The renal expression of gene HO-1 in control C57Bl6 mice was significantly decreased at 18 compared with 3 months of age [40]. In another study, protein HO-1 expression was not significantly different at baseline between 6–8-week-old mice and 1-year-old mice, but the older animals’ ability to upregulate HO-1 in response to ischemia reperfusion injuries (IRI) was impaired (especially in the medulla), and they exhibited worse renal function compared with young animals [41]. These results echo the renal sensitivity of elderly humans to nephrotoxic agents. 

To summarize, the preventive or genetic induction of HO-1 expression has been shown in many studies to confer renal protection. A decrease in renal HO-1 expression may contribute to the kidney’s sensitivity to certain pathologies. 

## 3. HO-1 in Renal Cell Populations

Comparative analyses have revealed that HO-1 is not expressed to the same extent in the distinct cell populations of the kidney for a given stimulus (Figure 1). During intravascular hemolysis or in the presence of HO-1-inducing stress stimuli or drugs in the circulation, the endothelial cells are the first cells to be challenged by these systemic stressors. If the stressor is filtered in the glomeruli, then the podocytes are exposed and may be damaged, as seen in diabetic nephropathy. Further, the stressor could encounter mesangial cells which have a secretory and immunomodulatory role and proliferate during HO-1 deficiency. Finally, the stressor may reach the tubular epithelium which are the most-studied cell type in relation to HO-1 (the proximal and distal tubules must be considered separately with respect to the epithelium). Tubular epithelial cells have the most potent capacity to overexpress HO-1 and suffer the most severe injury in case of its deficiency or during intravascular hemolysis and rhabdomyolysis. This section describes the key functions of the main kidney cell types and the specific role of HO-1 within them.

### 3.1. Endothelial Cells 

Endothelium is comprised of a monolayer of endothelial cells (EC) which accomplish critical functions in the blood vessels. They cover the interior of blood vessels and perform barrier functions, but they also serve as an exchange interface with adjacent tissues, a sensor for tissue and intravascular stress, and as a source of defense mediators [42]. As such, EC are the first line of cells to encounter circulating stressors and HO-1 expression is essential for their survival, with HO-1-deficient patients and murine models exhibiting extensive endothelial damage [19,43]. The pro-survival effects of HO-1 in EC occur in every organ and there exists a large body of work regarding its induction by various molecules [44,45]. According to its localization (in different organs, in vessels or capillaries), EC structure, specialization and functions vary. Thus, EC from different vascular beds have dynamic expression profiles (both spatially and temporally) and this might explain their selective involvement in different disease processes [42]. 

In the kidney, three main types of endothelium can be distinguished: glomerular, peritubular capillary, and medium/large vessel endothelium [46], each with different levels of HO-1 expression or function. Glomerular EC are “fenestrated” and covered by a glycocalyx [47] responsible for restricting macromolecule passage to the urinary chamber [48]. Despite the glycocalix, small hydrophobic molecules (e.g., heme) can enter cell membranes and activate proinflammatory and prothrombotic pathways, including TLR-4 signaling, thus promoting blood cell adhesion and vaso-occlusion (Schaer et al., 2013). Peritubular capillaries’ EC are also fenestrated and lay upon a thin stroma. They transport reabsorbed components, sustaining tubular epithelial cell function [46]. Finally, the endothelia of both medium and large vessels are contiguous, with interconnected EC. The structural and functional diversity of EC in the kidney as a function of their localization makes them differentially sensitive to specific stressors, such as toxic or proinflammatory filtrated molecules for glomerular EC, hypoxia for peritubular capillaries, or changes in shear stress, with correspondingly specific responses to such assaults [46].

The main factors affecting HO-1 expression in EC are hypoxia, shear stress, the presence of heme and other circulating stressors (ammonia [49], S-adenosyl methionine [50], etc.), circulating substances (statins [51], resveratrol [52], etc.) or drugs. Oxygen partial pressure decreases along the peritubular capillary, notably under stress conditions with excessive O_2_ consumption. Interestingly, HO-1 expression has been reported to decrease upon hypoxia while HO-2 expression was preserved in macrovascular human umbilical vein EC (HUVEC) [53]. However, HO-1 expression is inducible in microvascular peritubular capillaries’ EC under different conditions that may provoke hypoxia (heme, Angiotensin II, etc.). Physiological shear stress is a major determinant of EC survival, as shown in vivo in human carotid atherosclerotic plaques [54] and rabbit carotids with reduced blood flow [55]: it also induces the expression of anti-inflammatory factors (in vitro in HUVEC and in vivo in rabbit aorta [56]) and vasodilatory ones (in vitro and in vivo in mice [57]). Interestingly, HO-1 expression is induced by shear stress, in vitro in macrovascular (aorta) and microvascular (HMEC) human EC [58], and this expression was reported in rats to be directly related to the level of shear stress: higher expression at high flow levels (in arteries), but requiring other factors for HO-1 to be expressed at low flow levels (in microvessels) [59]. This phenomenon has so far only been reported in intestinal vessels. However, if these results are verified in other organs, it may explain decreased HO-1 induction in capillaries such as glomerular EC under stress [27]. EC are highly exposed to cytotoxic hemolysis-derived products and endothelial responses to these insults reflect the heterogeneity of EC. Microvascular EC, and particularly glomerular EC, are less prone to up-regulate HO-1 compared with macrovascular EC, both in vitro and in a mouse model of hemolysis [27]. This renders them particularly susceptible to injury through differences in their complement C3 regulation and heme degradation as compared with macrovascular EC [27]. Little is known about the HO-1 induction capacity of the peritubular capillaries’ EC. 

Renal macrovascular EC remain poorly studied with respect to HO-1 expression and function. The most widely used model in the study of macrovascular EC is the human umbilical vein EC (HUVEC). Exposure to heme in HUVEC is responsible for rapid signal transduction, mobilization of the endothelial Weibel Palade bodies, and NF-kB activation, thus conferring them a pro-inflammatory and pro-thrombotic phenotype [60,61,62]. However, longer exposure allows them to develop adaptation mechanisms and become highly resistant to oxidative-stress-mediated injuries and lipid peroxidation products. These include the induction of ferritin and HO-1 [63], though HO-1 induction in glomerular EC in vitro and in mouse models of hemolysis is much weaker [27,64]. HO-1 induction in macrovascular EC notably modulated complement activation [27,65], thrombomodulin expression in hemolysis conditions [27] and in the context of septic kidney injury [66], and expression of adhesion molecules associated with EC activation [67]. 

Taken together, these data indicate that the kidney endothelium is strongly influenced by toxins and stressors as a result of its filtration function. Such stressors seem to trigger strong adaptation mechanisms in macrovascular EC which allow them to resist insults to a certain level, mainly by the upregulation of HO-1 and its associated genes. While there is a marked lack of data in the literature, available studies suggest that, in contrast to macrovascular EC, this HO-1 upregulation is less pronounced in the glomerular microvascular endothelium, making it particularly sensitive to environmental stress. 

### 3.2. Podocytes

Podocytes are hyper-specialized cells that maintain the glomerular filtration barrier through the synthesis of glomerular basement membrane (GBM) components and the formation of the slit diaphragm by attaching their foot processes to the glomerular endothelial cells [68] (Figure 1). Their strategic localization, enabling their filtration function, makes them particularly sensitive to mechanical, oxidative and immunological stress. Podocytes have unique adaptive responses, including the upregulation of vimentin and desmin [69], or antioxidant proteins such as Sirtuin1 and metallothionein. Sirtuin upregulates Nrf2, thus enhancing HO-1 expression. [70,71] Interestingly, sirtuin-1 expression (together with Nrf2 and HO-1) has been reported as being increased by exposure to advanced glycation end products [71], and HO-1 was shown to be specifically enhanced in podocytes in diabetes patients [72]. Although less studied, the expression of HO-1 has also been described under hemolytic conditions [27,73], with heme inducing HO-1 expression in a Nrf2-dependent manner in podocytes. Interestingly, Nrf2-deficient mice spontaneously developed proteinuria with foot process effacement, decreased synaptopodin and nephrin expression, and podocyte apoptosis [73]. 

Though the literature on HO-1 in podocytes is not abundant, what evidence there is suggests a protective physiological role for the Nrf2/HO-1 axis by preserving pedicels and inhibiting the death and detachment of podocytes. In pathological contexts, HO-1 is overexpressed as a defense mechanism but could also be a byproduct of exhausted resistance mechanisms resulting from overwhelming stress, in which context it is insufficient to prevent podocyte injury. 

### 3.3. Mesangial Cells

Mesangial cells, together with the mesangial matrix they produce, form the mesangium, a support tissue for the glomerulus floculus. Among mesangial cells are immune cells, monocyte/macrophage-like (5 to 15%) and contractile cells, which are similar to smooth muscle cells (85 to 95%) and provide structural support and contraction [74,75]. Heme promotes the proliferation of smooth muscle cells [76] and its receptor TLR4 is expressed on mesangial cells [77], but the role of HO-1 in these cells is poorly studied. Nitric oxide (NO) has been shown to induce HO-1 in mesangial cells [78], HO-1-deficient patients have a mild mesangial proliferation [17,18,43], and HO-1 has been shown to modulate mesangial cell proliferation via p21 upregulation [79]. Further work is needed to determine whether the large body of knowledge accumulated for macrophages is applicable for the macrophage-like mesangial cells. 

### 3.4. Tubular Cells 

If HO-1 expression was described in tubules some time ago, it is only recently that its regulation has been described as being segment specific; HO-1 also plays distinct roles in different segments of the tubule in order to maintain renal functions [80]. 

#### 3.4.1. Proximal Tubule 

The proximal tubules are the cells in the kidney which show the highest capacity for the overexpression of HO-1. Indeed, proximal tubular epithelial cells have been shown to be especially sensitive to oxidative stress in vitro [81]. They rely strongly upon HO-1 for protection from, and adaptation to, stressors, as evidenced by the fact that tubular injury is the cardinal pathologic feature in human HO-1 deficiency [82]. The main function of the renal proximal tubule is uptake of water, ions, amino acids and filtered proteins. In the absence of glomerular damage, filtered proteins are mostly of a size below 68kDa. Internalization of proteins, including heme-associated ones such as myoglobin or hemoglobin, but also hemopexin, occurs via the megalin and tubulin receptors [83]. This reabsorption function requires a large amount of energy (ATP) provided by mitochondrial beta-oxidation [84]. However, severe ATP depletion results in mitochondrial injury, causing further depletion of energy stores and formation of reactive oxygen species [85]. This high metabolic demand produces a state of relative hypoxia in kidney medulla, and the proximal tubules are particularly sensitivity to ischemia/reperfusion and heme excess [86,87]. The sensitivity to heme is also explained by the constitutive expression of TLR4 (a heme receptor) by the proximal tubules [61], and TLR4 expression is increased in renal ischemia–reperfusion injury and septic injuries [77]. 

While HO-1 is expressed only weakly in the kidney under normal conditions, it is strongly inducible in proximal tubules under different forms of stress [88], proteinuria among them [80]. The interplay among uptake of heme-containing proteins, osmotic pressure driving heme influx, TLR4 expression and increased synthesis of heme in proximal tubules may together explain this specific, strong upregulation of HO-1. It has been suggested that HO-1 expression in tubular epithelial cells depends more upon basolateral exposure to heme (from the peritubular capillaries under hemolytic conditions) than intraluminal heme content. However, under conditions of hypoxia and of altered polarity of renal EC, HO-1 upregulation has been reported to depend more upon apical exposure to heme [89]. In tubular epithelial cells, HO-1 upregulation by heme is dependent on Nrf2 stabilization [90]. HO-1 has a dual role in tubular EC, being simultaneously an anti-oxidative and a regulator of cell death. Indeed, HO-1 is responsible for the upregulation of p21, a cyclin-dependent kinase inhibitor responsible for stopping cell cycle progression, which reduces apoptosis in tubular EC [91,92]. Furthermore, HO-1 expression decreases ferroptosis in proximal tubular cells, a non-apoptotic regulated cell death associated with accumulation of reactive oxygen species (ROS) derived from lipid peroxidation [93]. Finally, HO-1 has been shown to inhibit autophagy in proximal tubular cells [94].

Altogether, proximal tubular cells are massively exposed to HO-1 inducers in pathological contexts (filtration of hemoglobin, myoglobin, hemopexin-heme complexes; ischemia, filtration of toxic products or drugs, etc.) and rely upon HO-1 for their protection, but HO-1 is also a marker of tubular stress and injury when their capacity for adaptation is overwhelmed. 

#### 3.4.2. Distal Tubule

Distal tubular EC along the medulla, notably in the thick, ascending limb of the loop of Henle (mTAL) and the distal convoluted tubule (DCT), are involved in the concentration and dilution of urine and the maintenance of homeostatic salt and solute levels by expressing different channels specific to their localization and cell subtype. During water reabsorption, the concentration of toxins (notably heme) becomes much higher in the distal nephron segments. Because of the low partial pressure of oxygen along the peritubular capillaries, these distal tubular EC are particularly vulnerable to hypoxia and rely more on anaerobic, glycolytic ATP production than proximal cells. Distal tubules have a greater ability to survive and adapt to hypoxic stress [95], being less sensitive to cell death, especially after ischemic injury [96]. 

Distal tubules, like proximal tubules, only weakly express HO-1 at a basal level; upon stimulation, however, its induction by heme has been called into question [88]. (Interestingly, HO-2 may be enhanced by heme in distal tubules [97].) However, HO-1’s role in distal tubules remains to be fully elucidated as its expression has also been associated with improved renal function after ischemia/reperfusion in cadaveric donor transplantation [98], as well as in CKD [99]. This may be related to the regeneration function of distal tubular epithelial cells, but further studies are needed to better describe the relationship between HO-1 and distal tubular epithelial cells’ functions.

To conclude, while data regarding distal tubules are lacking, some evidence suggests that enhanced HO-1 expression could confer protection to distal tubules’ EC. 

## 4. HO-1 in Kidney Diseases

HO-1 and its induction in various kidney pathologies and physiological conditions aroused interest twenty years ago, particularly in the context of massive free heme release, such as that associated with hemolytic diseases and rhabdomyolysis [100]. However, HO-1 has exhibited a protective role in many other pathologies that are not classically considered as heme-mediated nephropathies. This encompasses ischemia/reperfusion renal injuries and tubular nephropathies (which may be toxic), septic and obstructive nephropathies and, moreover, glomerular nephropathies. This classification of disease represents a spectrum of different contributions from HO-1 stimulating stressors (Figure 3). Indeed, ischemia followed by reperfusion can be part of the mechanism of renal injuries in hemolytic pathologies [101] and rhabdomyolysis [102]. On the other hand, local heme release was recently reported in experimental ischemia-reperfusion [103], and may be a contributing factor in ischemia/reperfusion-related pathologies such as renal transplantation, though without it being the main damaging factor [104]. Furthermore, high plasma levels of free heme have been measured in some pathologies not classically considered as being heme-related, such as sepsis [105]. Having said this, we will consider the two extreme parts of the spectrum to illustrate the protective role of HO-1 in both the presence and absence of intrarenal heme overload.

### 4.1. Diseases Associated with a Massive Release of Free Heme 

Two broad categories of pathology are associated with massive releases of heme, namely, hemolytic diseases and rhabdomyolysis. In hemolytic diseases, heme is released from hemoglobin after rupture of the red blood cells. The oxidation of Hb and liberation of heme can occur in the circulation or in the tubules, after filtration of the Hb dimers in the glomeruli. During rhabdomyolysis, myoglobin from injured skeletal muscle fibers enters the circulation, but heme is not released there due to its higher affinity for this protein compared to Hb. Instead, myoglobin is filtered by glomeruli and heme is released in the interstitium, especially after intake of this protein by the proximal tubules. In both cases, HO-1 is massively upregulated in the kidney. Using several examples, we will illustrate and analyze the role of HO-1 in diseases associated with a massive release of heme.

#### 4.1.1. Hemolytic Diseases

Hemolysis is a pathological condition encompassing various etiologies which can occur either intra- or extravascularly. Intravascular hemolysis is the destruction of red blood cells (RBCs) in the circulation, which may be a complication of extravascular hemolysis, or else occur mechanically such as in thrombotic microangiopathy, malignant hypertension or from mechanical heart valve implants [28]. Extravascular hemolysis is the elimination of the RBCs in the spleen, liver or bone marrow by phagocytes. It is mainly linked to defects which may be intrinsic (including genetic diseases such as sickle cell anemia or acquired diseases such as paroxysmal nocturnal hemoglobinuria (PNH)) or extrinsic (such as autoimmune hemolytic anemia, hepatic cirrhosis). In both cases, lysed RBCs release hemoglobin, the auto-oxidization of which liberates free heme which possesses pro-oxidative and pro-inflammatory properties. Haptoglobin and hemopexin in the circulation scavenge hemoglobin and heme, respectively, and drive their elimination. However, these protective mechanisms are rapidly exhausted and cannot prevent hemoglobin and heme causing harm during a massive hemolytic crisis. Indeed, free heme levels have been shown to be elevated in the plasma of sickle cell disease (SCD) patients (4–50 µmol) [106,107,108] and a mouse model of SCD (75–120 μM) [109]. The pro-inflammatory effects of the remaining free heme and free hemoglobin may provoke organ damage, including of the kidney [110]. Notably, hemoglobin cast nephropathy has been reported in kidney biopsies of patients with hemolysis with various etiologies, such as PNH, autoimmune hemolytic anemia, hemoglobinopathies and transfusion accidents [111]. Regarding free heme that escapes hemopexin scavenging, only HO can degrade it and this detoxifying step is critical during hemolysis, as shown during hemoglobin infusion in HO-1-deficient mice which exhibited acute renal failure and marked mortality [112]. However, before describing the involvement of HO-1 in specific hemolytic pathologies, we will first examine the main results from a validated and widely used mouse model of intravascular hemolysis, namely, phenylhydrazine (PHZ) treatment. Thereafter, we will focus on the main hemolysis etiologies associated with kidney injuries and variations in HO-1 expression, namely: sickle cell anemia, hemolytic uremic syndrome, malaria and PNH.

##### PHZ-Induced Hemolysis Model

The most-used in vivo model for intravascular hemolysis employs an injection of the hemolytic drug PHZ. It induces massive RBC destruction via membrane lipid peroxidation [113] and destabilization of hemoglobin. Proposed as a treatment in the mid-20^th^ century for patients with polycythemia (without much success), it has been used in over 200 papers to study hemolysis in experimental animals. PHZ-treated mice exhibit an increased tubular expression of HO-1 [114,115], which can be prevented by administration of Zinc Protoporphyrin (ZnPP) [116]. This upregulation is thought to be Nrf2 dependent since Nrf2-deficient mice exhibit a more severe, hemolysis-induced AKI phenotype, enhanced tubular injury markers (KIM-1 and NGAL) and cell death, as well as lower HO-1 and ferritin [117]. However, other studies suggest that despite elevated HO-1 expression, renal alterations in this model are at least partly independent of heme, since injection of free heme could not reproduce them and administration of the heme scavenger hemopexin could not prevent them in the PHZ model [115]. If complement overactivation in PHZ-induced hemolysis is involved in tissue damage [118], HO-1 expression seems to be independent of it, despite the kidney injury phenotype, since expression of HO-1 was comparable in wild type (WT) and C3^−/−^, PHZ-treated [118] and factor H (FH)-pretreated mice [119]. Therefore, HO-1 expression appears to be dependent on hemolytic status rather than complement-mediated tubular injury [119], and hemopexin prevented complement activation in the PHZ model without affecting HO-1 expression [120]. Thus HO-1 in the PHZ model protects the kidney from heme-dependent renal alterations, probably without influencing complement activation or other heme-independent injuries. 

##### Sickle Cell Anemia

Sickle cell anemia (SCA) is a monogenic disease, characterized by a mutated Hb gene (HbS), which can polymerize and thus deform erythrocytes, reducing their lifespan [101]. Local hypoxia in a restricted area, for example, during a vaso-occlusive crisis, increases their polymerization capacity and instability and induces massive, intravascular hemolysis which can alter renal structure and function [121,122]. Both SCA and its heterozygous state (sickle cell trait, with an attenuated phenotype), are associated with an increased risk of CKD [123]. In addition to sickle cell susceptibility to hemolysis, HbS is unstable and prone to auto-oxidation and release of its cofactor heme. Kidney injuries in SCA are described as the result of systemic endothelial vasculopathy leading to impaired renal hemodynamics and accompanied by hyper glomerular filtration and hyperfunction of the proximal tubules. These features are associated with glomerular injuries such as Focal Segmental Glomerulosclerosis (FSGS) and interstitial chronic inflammation (secondary to medullary hypoperfusion), both of which are responsible for CKD [121]. HO-1 expression is increased in renal tubules, interstitial cells and in the vasculature of biopsies from SCA patients compared with normal kidneys [124]. However, HO-1 is not upregulated in mesangial cells, despite frequent mesangial cell proliferation in SCA [125], suggesting it could be mediated directly by heme as discussed above [76]. Interestingly, long GT tandem repeats, responsible for decreased inducibility of HO-1 expression [126], were independently associated with the occurrence of AKI in a monocentric SCA patients cohort [127], raising the possibility of a protective effect of HO-1 expression in patients; HO-1 expression is also induced in kidneys of various SCA mice models [124,128,129]. This murine expression seems to be relatively organ specific: compared with the control mice (Townes-AA), kidney and liver HO-1 expression was 10-fold greater in Townes SS mice (a validated SCD model), but only 3- to 4-fold greater in the heart and spleen [128]. In addition, in a global transcriptional analysis of SCA patients, HO-1 and its by-products were upregulated in circulating blood mononuclear cells together with genes involved in heme metabolism, cell-cycle regulation, antioxidant and stress responses, inflammation and angiogenesis [130], which suggests that circulating mononuclear cells could participate in compensating for sickle cell vascular injury. SCD patients displayed an increased population of circulating monocytes expressing high levels of HO-1 with anti-inflammatory properties (37% vs. 7% in controls), and could be involved in removing hemolysis-damaged endothelial cells, thus confirming the key role of circulant cells expressing HO-1 [131]. Interestingly, CO (one of the by-products of heme degradation by HO-1) has been suggested as being involved in hyperperfusion and hyperfiltration (both observed in the kidney of SCA patients [121,132,133]), and may thus counterbalance the vasoconstrictive effects of NO depletion by hemoglobin scavenging [108], as well as vasa recta rarefaction in medulla [134]. 

Microparticles (MPs—enriched in heme in SCA) are also involved in kidney pathology by inducing endothelial injury and facilitating acute, vaso-occlusive events in transgenic SAD mice [129]; they have also been shown to upregulate HO-1 expression in human EC [135]. Finally, recent studies have identified a key role for complement overactivation in the tissue damage seen in SCD [115,136,137], but as discussed in relation to the PHZ model, HO-1 expression seems to be independent of complement activation. Another model of mouse SCD reportedly showed decreased vascular congestion and down-regulation of injury-related genes in comparison with untreated mice (the SCD model had β-globin replaced by transgenes for human βS and βS-antilles globins on a C57Bl6 background, and was treated chronically with SnPP, a short-term HO-1 inhibitor which also induces HO-1 expression over the longer term [138]). This reflects the lack of specificity of SnPP, which has already been suggested for use against another protoporphyrin [139,140]. In conclusion, there is evidence for both a local and systemic (through circulating cells) protective effect of HO-1 on both kidney hemodynamics and levels of pro-oxidant stress in SCA-associated kidney injuries. 

Thus, HO-1 has been implicated in SCA with respect to the preservation of local kidney hemodynamics and the protection of endothelial and tubular cells against oxidative stress. 

##### Hemolytic Uremic Syndrome

Hemolytic uremic syndrome (HUS) is a rare, kidney-predominant, thrombotic microangiopathy (TMA) related to a dysregulation of the complement alternative pathway (AP—“atypical” or aHUS) and/or the presence of a Shiga-like toxin, secreted by a pathogen (“typical” or STEC-HUS) [141]. The formation of fibrin-platelet clots in the glomerular microvessels, promoted by endothelial pro-inflammatory activation, leads to intrarenal mechanical hemolysis. Genetic or acquired complement dysregulation in aHUS, or expression of Shiga-like toxin by a pathogen in STEC-HUS, nevertheless, does not fully explain the occurrence of these pathologies as its penetrance is incomplete in both cases and interest in heme as an additional factor in endothelial damage has increased [60]. Tubular HO-1 staining by immunohistochemistry was reported in kidney biopsies from two aHUS patients, at a proportional level to the degree of hemolysis; no glomerular endothelial HO-1 staining was observed, while positive staining was detected on podocytes [27]. Until recently in typical HUS, despite the hemolysis in this pathology suggesting involvement of heme and HO-1, data have been scarce. Only one article suggested that Shiga-like toxin (Stx), secreted by *E. Coli*, could inhibit HO-1 expression in renal epithelial cells [142]. However, a recent article reported significantly elevated levels of plasma heme in STEC-HUS patients compared with controls, inversely correlated with low plasma hemopexin levels but significantly associated with elevated plasma levels of HO-1 [64]. Moreover, microvascular glomerular EC incubated with heme upregulated HO-1 expression independent of the presence of STx. 

The capacity of glomerular EC to upregulate HO-1 is weaker than other EC types [27]; however, the decreased inducibility of HO-1 in glomerular EC could explain their specific susceptibility in HUS.

##### Malaria

Infection with the parasite *Plasmodium* is responsible for extensive hemolysis with hemoglobin and free heme release. Acute kidney injury is a clinical hallmark of disease severity and malaria is also associated with collapsing glomerulopathy [143]. Labile heme concentrations are increased in the plasma of patients and in mouse models; interestingly, labile heme levels in the latter increased more strongly in urine compared to plasma [144]. HO-1 expression was shown to be protective in a cerebral malaria mice model of *P. berghei* ANKA via CO generation which, because it binds to hemoglobin, prevents its oxidation and the generation of free heme [145]. Malaria was associated with HO-1 upregulation in proximal tubular epithelial cells in the same model, as well as in a *Plasmodium chabaudi chabaudi* mice model [144,146]. In the latter model, HO-1^−/−^ mice had extensive proximal tubular necrosis with hemoglobin casts and the beneficial effects of HO-1 expression for survival were dependent on its specific proximal tubular location (as opposed to hepatic, endothelial or cerebral expression) and was Nrf2-dependant [144]. 

HO-1 upregulation in tubular epithelial cells thus appears to be reno-protective in the context of malaria.

##### Paroxysmal Nocturne Hemoglobinuria 

Paroxysmal nocturne hemoglobinuria (PNH) is an acquired clonal defect of phosphatidylinositol glycan anchor synthesis, responsible for enhanced RBC susceptibility to hemolysis [147]. PNH has been associated not only with AKI [148,149,150,151,152] but also with CKD (risk increased >6 fold) [153]. Interestingly, complement inhibition by Eculizumab improves renal function in all stages of CKD (within 6 months). The mechanism proposed for such improvement was the restoration of NO availability and subsequent changes in renal vascular tone. Increased expression of HO-1 has been reported in the cortex and outer medulla tubular EC and macrophages (notable in hemosiderin rich areas) [154,155]; in vitro, enhanced expression of HO-1 was observed when renal tubular epithelial cells were exposed to PNH patients’ urine [155]. However, the role of this enhanced HO-1 expression in PNH patients’ kidneys requires further study, notably, for example, in correlation with clinical data regarding kidney function and the effects of complement inhibition.

#### 4.1.2. Rhabdomyolysis 

Rhabdomyolysis is a massive skeletal muscle injury of diverse etiology, notably trauma (“crush syndrome”), muscle hypoxia, genetic defects, medication or drug abuse [102]. This destruction is responsible for excessive release into the circulation of the intracellular content of muscle fibers, including electrolytes, myoglobin, and other sarcoplasmic proteins (creatine kinase, aldolase, etc.). Myoglobin and its oxygen carrier, heme, are particularly involved in the most life-threatening complication of rhabdomyolysis, acute kidney injury (RIAKI), responsible for a significant morbi-mortality in intensive care units [156]. The mechanism for kidney injuries is classically described as an association of renal vasoconstriction [157], proximal tubular aggression (by oxidative stress, lipid peroxidation [158,159] and macrophage activation [160,161]) and intra-tubular precipitation of myoglobin. Recently, complement has also been reported to play a contributory role—in a partly heme-dependent manner—in RIAKI [30,162].

Induction of HO-1 expression was discovered nearly 30 years ago in a rat model of glycerol-induced rhabdomyolysis [34]. In this model, a single infusion of hemoglobin prior to glycerol challenge enhanced HO-1 expression sufficiently to prevent kidney failure and mortality was drastically reduced (from 100% to 14%). On the contrary, competitive inhibition of HO-1 appears to increase AKI: in the same glycerol mice model, HO-1 ^−/−^ mice had fulminant, irreversible renal failure and 100% mortality [112]. HO-1 expression increased to a maximum 48h after glycerol injection and then decreased, whereupon a concomitant decrease in protective anti-inflammatory effects was observed [163]. HO-1 expression in kidney in response to glycerol challenge is particularly pronounced in comparison with other models of AKI (such as ureteral obstruction, cisplatin, ischemic/reperfusion injury), as plasmatic concentration and intrarenal HO-1 protein and mRNA reached maximal levels in this model [164,165]. Upregulation of HO-1 expression by pyruvate (a hydrogen peroxide scavenger) prior to glycerol injection was associated with renal protection against RIAKI [166], but also increased renal glucose concentrations and IL-10 mRNAs, while reducing monocyte chemoattractant protein 1(MCP-1) and tumor necrosis factor α (TNF-a) [167]. This suggests an immune-modulating role of HO-1 in RIAKI. However, the cytoprotective role for HO-1 in the kidney is insufficient for its protection against RIAKI; for example, despite significantly higher expression of HO-1, mice lacking heavy-chain ferritin had a more severe phenotype after glycerol injection [168]. HO-1 could thus play a passive, bystander role since HO-1 expression has been shown to be upregulated simultaneously with ferritin [34]. On the other hand, HO-1 is not alone in affording cytoprotection in glycerol-induced RIAKI, as N-acetylcystein treatment (a scavenger of free radicals) also showed beneficial effects [169]. 

In addition, preventive HO-1 induction, together with the induction of other anti-inflammatory and cytoprotective stress proteins such as ferritin, haptoglobin, hemopexin, alpha 1 antitrypsin and IL-10, has been suggested to procure protective effects, notably in a glycerol-induced RIAKI model. This has often been called a “preconditioning treatment” in the literature. For example, infusion of nitrated myoglobin or SnPP played a synergistic role in HO-1 induction, thus preserving renal function [170]. The anticholinergic agent Penehyclidine hydrochloride (PHC) [171], but also curcumin (a polyphenol with antioxidant, anti-apoptotic and anti-inflammatory properties) [172,173], have both been shown to exert a preventive effect on glycerol-induced AKI while also decreasing cell apoptosis in renal tissue, enhancing Nrf2 activation and HO-1 induction. 

Interestingly, the heme receptor TLR4 may not be implicated in the effects observed, since TLR4^−/−^ mice were not protected against glycerol-induced RIAKI, and the TLR4 inhibitor TAK-242 failed to preserve kidney function despite preserving blood flow [174]. Furthermore, HO-1 expression in tubular cells in vitro was not inhibited by TAK-242; this suggests that other heme receptors are involved in HO-1 induction. Receptor for advanced glycation end-products (RAGE) has also been confirmed as a receptor for heme [175] and is expressed in tubular cells [176]—it will be interesting to explore its role in RIAKI. Expression of HO-1 was reported to be independent of complement system activation in RIAKI, since HO-1 was upregulated equally in both wild type and C3^−/−^ mice injected with glycerol, despite reduced kidney lesions in C3^−/−^ mice [30].

Finally, HO-1 has a protective role mainly by improving renal microcirculation, reducing oxidative stress and probably also by regulating the immune response induced in RIAKI. This protective role is, however, insufficient by itself to completely protect the kidney from RIAKI lesions.

### 4.2. Diseases without Massive Cell-Free Heme Release

This category encompasses several pathologies affecting the kidney which are not classically associated with massive heme release, and, in accordance with the aim of this review, we will present the numerous pathologies in which HO-1’s role has elicited research interest. Our aim is to comprehensively examine how HO-1 expression is involved with the physiopathology of diseases with little in common except kidney damage rather than to be exhaustive. Indeed, HO-1 has been involved in toxicity (gentamycin, cisplatin, calcineurin inhibitor, radiocontrast product, mercuric chloride), septic, mechanic (obstruction), hemodynamic (hypertension, ischemia-reperfusion), metabolic (diabetes), genetic (Polycystic Kidney Disease (PKD)) and renal-specific (membranous nephropathies, anti GBM) and systemic (Lupus nephropathy) immunologic pathologies (Table 1). Moreover, these diseases affect podocytes, mesangial, tubular epithelial, endothelial and interstitial cells in a variety of ways, thus confirming the pleiotropic effects of HO-1 in both physiology and disease. Finally, HO-1 has been involved in the acute phase of kidney injury, but also in the chronic phases of renal damage. We chose to focus on deciphering HO-1’s roles in pathologies that are the subject of a substantial body of literature or the object of clinical trial(s) regarding HO-1 (with the exception of preeclampsia and PKD, for which data remains scarce). The protective functions of HO-1 in other pathologies, which from the literature could be equally important as those which are currently the subject of clinical trials, will be summarized in Table 1 and future therapeutic prospects regarding these will be discussed in the next section.

#### 4.2.1. Ischemia/Reperfusion 

Ischemia/reperfusion (I/R) injury is a result of an inflammatory process, triggered by transient reduction or interruption of blood flow in an organ and resulting in oxygen and nutrient deprivation, followed by a reperfusion. ATP depletion is responsible for cell injury and, if severe enough, cell death by necrosis or apoptosis [85]. Among relevant etiologies are embolic or thrombotic events, circulatory “shock” and hypotension, cardiac and vascular surgery and transplantation [289]

I/R can be induced in the kidney by unilateral or bilateral renal arterial clamping, which have been extensively studied in both mouse and rat models. In such models, HO-1 expression was notably increased in tubules [228]. HO-1 ^−/−^ mice exhibited severe renal dysfunction and IL-6-mediated mortality following ischemia, suggesting a protective role for its upregulation in I/R [229]. Pharmacological inhibition of HO-1 by tin mesoporphyrin (SnMP) was also responsible for renal failure [228]. On the contrary, tin protorporhyrin (SnPP, which has an immediate HO-1 inhibition effect, but induces HO-1 after a few hours) prevented injury when injected at a high concentration 24 h before I/R [230]. In addition, HO-1’s effects upon increased resistance to apoptosis in human tubular cells may be involved in protection against tubular injury [91]. However, tubular cells are not the only cell type to be involved, as ischemic injury in HO-1^−/−^ mice also resulted in a significant increase in activated renal macrophages and neutrophils and a simultaneous decrease in intrarenal, dendritic resident cells [231]. Furthermore, upregulation of HO-1 has been shown to protect kidneys from ischemic renal failure [232], and mice with myeloid-restricted deletion of HO-1 exhibited significant renal damage, pro-inflammatory responses and oxidative stress 24 h after reperfusion, but also an impaired tubular repair 7 days after I/R [233]. Thus, HO-1’s upregulation also seems to have a role in renal protection through myeloid circulating cells, notably macrophages, in addition to tubular cells.

HO-1 inducers have generated significant interest in animal models of I/R and many of them have shown protective effects. Among them, hemin exhibited conflicting effects, probably owing to the timing and concentration of administration, namely, protective at low doses and before ischemic injury but deleterious in other cases [230,234]. HO-1 upregulation by Cobalt-Chloride (Co-Cl2) was associated with decreased basal NO concentration, and even inhibited the ischemic increase in NO while preserving post-ischemic glomerular filtration rate (GFR) and blood flow in the outer medullar. These effects where deleted when HO-1 was inhibited pharmacologically by SnMP [235]. Interestingly, statins’ anti-inflammatory effect on I/R injuries could be mediated by upregulation of HO-1 in circulating monocytes/macrophages [236]. Some HO-1 inducers with beneficial effects on I/R (adiponectin [290], bardoloxone methyl [205]) were shown to be Nrf2-independent, acting instead through a peroxisome-proliferator-activated-receptor-α (PPARα) which recognizes a specific region of the HO-1 promoter. Other studies suggested that NGAL (a tissue-specific marker of injury) and the hypoxia inducible factor (HIF) 1 alpha could also play a role in HO-1 upregulation in I/R [237,238]. 

Prevention of I/R injuries is a serious challenge during cardiac or vascular surgery. In patients, longer GT repeats in the HMOX1 gene promoter (leading to reduced HO-1 inducibility) is associated with an increased risk of AKI after cardiac surgery [38], hinting at a link with HO-1 expression. Indeed, ischemic preconditioning (by vascular clamping) showed promising protective effects in animal models of I/R and the potential implication of HO-1 in the observed effects was suggested [239,240], since a priori augmentation of the expression of HO-1 induces resistance to apoptosis in human tubular cells [91]. However, HO-1 expression was not increased in such models [241,242] and the results of such procedures on AKI incidence in human cardiac surgery were also contradictory [243,244].

Transplantation is, by definition, an I/R situation whose severity varies depending on the kind of the donor (living, brain dead or circulatory dead donor), the graft characteristics (extended criteria or not), but also on cold and warm ischemia times which can lead to delayed graft function or primary non-function. Lower expression of *HO-1*, *VEGF165*, and *Bcl-2* has been evidenced in kidneys from cadaveric donors in comparison with living donors [98], but no difference was observed between extended criteria donors compared with normal donors [245]. In a rat model of transplantation, induction of HO-1 by hemin [246], by hyperthermic preconditioning [247,248] or by administration of cobalt protoporhyrin (CoPP) [248] or Fenoldopam [249] in the donor were all associated with preserved kidney graft function and prevention of post-reperfusion apoptosis. Duration of warm ischemia time is associated with delayed graft function and has a profound impact on I/R injuries. Indeed, long (45 min) but not short (15 min) warm ischemia induced inflammation, local tissue injury, higher cell-free heme concentrations and increases in HO-1 but also C5aR, IL-6 and TNF-α expression [103]. If cold ischemia time is also associated with delayed graft function, HO-1 expression during this period is probably a bystander effect of the preceding warm ischemia time, though its role has been poorly studied [250]. Interestingly, HO-1 also plays a determining role in transplantation, besides this major protective role in I/R, by preventing Rapamycine toxicity [221], acute rejection (affecting the recipient cellular response and by protecting the graft itself) [251,252] and improving long-term graft survival [256]. Genetic polymorphisms in the HO-1 promoter have been suggested to be associated with better kidney function in the receiver, but with conflicting results [39,253,254,255], and HO-1 has generated interest in xenotransplantation [257,258].

To conclude, HO-1 upregulation during I/R in tubular cells has an essential protective effect, notably by modulating local hemodynamics through NO synthesis regulation, but HO-1 expression by myeloid cells is also key in preserving kidneys from pro-inflammatory effects of recruited circulating immune cells. Many therapeutic molecules have been tested in animals, providing new insights into HO-1 regulation and promising results that could improve clinical practice, in particular for interventions that are (at least partly) scheduled, such as cardiovascular surgery or transplantation. 

#### 4.2.2. Sepsis 

Sepsis-associated AKI (S-AKI) is a frequent complication in critically ill patients and is associated with morbidity and mortality [291]. Sepsis has a complex and unique pathophysiology, incompletely understood, that distinguishes it from other AKI phenotypes. If S-AKI was previously considered to be merely the natural evolution from decreased renal perfusion to tubular epithelial cell death (or acute tubular necrosis), the broader modern interpretation includes microvascular dysfunction, inflammation, and metabolic reprogramming [291,292]. Resistance and tolerance have been suggested as two different mechanisms that preserve kidney function and sustain wound healing during sepsis [293]. Lipopolysaccharide (LPS) has been known to induce HO-1 expression for almost 50 years [294]. It has been implied that HO-1 confers specific kidney resistance against LPS because HO-1^−/−^ mice exhibited a greater reduction in GFR and renal blood flow, increased renal cytokine expression, and increased activation of NF-κB after LPS injection [279]. In a model of polymicrobial sepsis (cecal ligation and puncture), HO-1-deficient mice displayed a higher mortality rate [280], but also increased Blood Urea Nitrogen (BUN) levels and tubular epithelial necrosis [105]. Levels of plasmatic, cell-free heme were elevated after high-grade infection while hemopexin treatment reduced mortality and improved kidney function (notably improved BUN levels), thereby suggesting a heme-dependent mechanism being responsible for the observed results [105]. Enhanced HO-1 expression, and in particular its capacity for CO formation in macrophages, was reported as essential for pathogen clearance and as contributing to renal protection against AKI [281]. 

Taken together, HO-1 expression, which is enhanced by endotoxins and probably by elevated heme levels during sepsis, is critical in renal resistance to pathogen-induced tubular epithelial necrosis but also more generally for the clearance of pathogens at the whole-body level. 

#### 4.2.3. Hypertensive Nephropathy

Hypertension is the first or second primary cause of progressive CKD (depending on geographic location) and is one of the most important and frequent causes of renal complications. During hypertension, the role of Angiotensin II (Ang II) is central, besides its hemodynamic role, and has been associated in animal models with podocyte loss [295], enhancement of matrix production by mesangial cells [296] and glomerulosclerosis, but also with changes in transporter expression in the tubule [297] and interstitial fibrosis [298], as well as oxidative stress and inflammation [299]. 

In animals, HO-1^−/−^ mice reportedly had the same blood pressure as controls suggesting HO-1 is not involved in the maintenance of blood pressure under physiological conditions [261]; nevertheless, unilateral kidney clamping in HO-1^−/−^ animals induced more severe hypertension and cardiac hypertrophy than in heterozygous or HO-1^+/+^ mice, with extensive ischemic injury at the corticomedullary junction [261]. Infusion of Angiotensin II (Ang II) increased HO-1 mRNA levels and its expression in adventitial and endothelial cells in Ang II–induced, hypertensive rat aortas, but losartan or hydralazine treatments prevented both Ang II induction of hypertension and upregulation of HO-1 expression [262]. Overexpression of HO-1, and subsequent increased generation of CO, was shown to reduce the effects of Ang II on blood pressure [263]. HO-1 ^−/−^ mice also had increased blood pressure compared with HO ^+/+^ after Ang II infusion, and this was associated with endothelial inflammation in vivo, and with aortic infiltration of pro-inflammatory monocytes and neutrophils [264]. The effect of Ang II on HO-1 have been suggested to be mediated by the Angiotensin Receptor 1 (AT1) [265].

Heme administration, which increases renal HO-1 expression, decreased blood pressure in spontaneously hypertensive rats, both when coupled with administration of arginate [266] or on its own [267]. The effects of heme on blood pressure were counterbalanced by administration of the HO-1 inhibitor ZnPP [267]. 

The medullar expression of HO-1 has been iteratively explored: indeed, medullary interstitial infusion of the HO-1 inhibitor zinc deuteroporphyrin 2,4-bis glycol (ZnDPBG – by an intramedullary dialysis probe) decreased medullar blood flow and suggested an important role for HO-1 in its maintenance [25]. On the other hand, medullary infusion of CoPP, a potent HO-1 inductor, preserved uni-nephrectomized mice from Ang II–dependent hypertension [268]. Several models of HO-1-specific overexpression in TALH were developed and have indicated a specific role for HO-1 regarding the protection of TALH cells from Ang II (notably decreased prostaglandin E_2_ (PGE_2_) levels and Ang II-induced DNA damage) [269] and in decreasing blood pressure, but also in reducing medullary Na-K-Cl cotransporter 2 (NKCC2) transporter expression and sensibility to furosemide [270], thereby modulating natriuresis as previously suggested [271].

In conclusion, if HO-1 is not essential to preserve arterial blood pressure in physiological conditions, it nevertheless has a critical role in reducing blood pressure under pathological conditions, notably under increases in Ang II. Furthermore, medullar and, notably, TALH expression of HO-1 seems to influence many factors implicated in chronic vascular nephropathy, such as decreased blood flow and natriuresis, thus making it highly eligible as a target for future therapeutics.

#### 4.2.4. Diabetic Nephropathy

Diabetic nephropathy (DN), which is a major microangiopathic complication of diabetes, is described as the result of chronic hyperglycemia and accumulation of the resultant advanced glycated products [300], podocyte loss [301], enhanced oxidative stress [302] and impaired microcirculation [303], orchestrated notably by innate immunity [304] leading to fibrosis, matrix deposition, and progressive renal injury [305].

Glomerular HO-1 expression, which is weak in the majority of renal diseases, has unexpectedly been reported as increased in different animal models of diabetes such as streptozotocin (STZ)-induced diabetes mellitus [72,187,188], and in a genetic model of diabetic db/db mice [189]. Even partial genetic deficiency in HO-1 is sufficient to sensitize mice to the development of diabetic glomerular microvascular lesions [190]. In vitro, high-glucose concentrations increased HO-1 expression in podocytes, and also their apoptosis. HO-1 inhibition (by ZnPP or HO-1 siRNA) increases the number of apoptotic podocytes both in vitro and in vivo, again suggesting a protective role for HO-1 [72]. The effects of HO-1 induction by CoPP on apoptosis and glomerular injury were confirmed in a model of STZ-treated, spontaneously hypertensive rats (SHR), who exhibited decreased NF-κB-induced inflammation and oxidative stress [191].

The protective role of HO-1 in DN affects not only podocytes but also mesangial cells, via the influence of Nrf2 [192]. Mouse mesangial cells overexpressing Nrf2 and exposed to high glucose showed upregulated expression of HO-1, as well as decreased ROS production and cell proliferation. These observations were confirmed by experiments with Nrf2-knockout mice in which the opposite was reported. 

Enhanced HO-1 expression also plays a role in renal hemodynamics, as shown in STZ-treated rats receiving CoPP, through upregulation of extracellular superoxide dismutase (SOD) and endothelial Nitric Oxid Synthase (NOS) leading to endothelial relaxation and decreased ROS production [193]. Furthermore, HO-1 inhibition by SnMP increased renal vascular resistance and altered GFR and renal blood flow, and this effect was reversed by co-administration of an SOD mimetic or a CO-releasing molecule [194]. 

Finally, HO-1 also plays a role in insulin sensitivity: its enhancement may reduce adipose tissue volume and cause remodeling of adipose tissue in a diabetic rat model of obesity-induced insulin resistance [195]. In type 2 diabetes patients with poor glycemic control, T(-413)A SNP in the HO-1 promoter was significantly associated with albuminuria development [196], but larger studies are required to confirm these results.

Among HO-1 inducers, hemin has been closely examined in co-administration with STZ and was associated with drastically increased HO-1 expression, especially in tubules, which interestingly was associated with improved renal function, decreased inducible NOS (iNOS), blood glucose levels, micro-albuminuria, glomerulosclerosis and fibrosis at 60 days [197]. These observations further suggest a crosstalk between glomeruli and tubules regarding HO-1. Hemin has been proposed to act through selective enhancement of an anti-inflammatory macrophage-M2 phenotype and IL-10, following a study inhibiting pro-inflammatory macrophage-M1 phenotype and suppressing the extracellular matrix/profibrotic factors responsible for renal lesions and interstitial macrophage infiltration in STZ ND rats [198]. Hemin could also exert protective functions on endothelium through synthesis of bilirubin, as proposed in the db/db genetic mouse model of diabetes where hemin and bilirubin were reportedly enhanced, similar to the endothelium-mediated relaxation of aortas [189]. Several other HO-1 inducers, namely, sinapic acid [199], artemisin [200] and puerarin [201], reportedly reduced kidney injuries in STZ-treated rodents, offering up the possibility of new therapeutic approaches.

As mentioned above, HO-1 is not the only HO implicated in the protection against diabetic nephropathy, and HO-2 deficiency has been shown to enhance STZ-induced renal dysfunction and injury, while HO-1 upregulation prevented these negative effects of DN [306].

To conclude, HO-1 plays a multifaceted and unexpected role in DN by limiting apoptosis of podocytes and mesangial proliferation, modulating renal hemodynamics and bolstering insulin sensitivity, but it may also influence glomerulo-tubular crosstalk in a heme-dependent manner.

#### 4.2.5. Chronic Kidney Disease 

While HO-1 has pleiotropic roles in specific nephropathies, these do not add up to the totality of known kidney pathologies. Indeed, each of the pathologies described previously in this section, as well as numerous others not mentioned here, can lead to progressive and permanent death of renal tissue and the replacement of functional nephrons with interstitial fibrosis characteristic of chronic kidney disease (CKD). Unilateral ureteral obstruction (UUO) is the gold standard rodent model for studying rapid, progressive interstitial fibrosis [272]. Early HO-1 expression has been observed in the peri-glomerular and peritubular interstitium [273]. Interestingly, preventive enhancement of HO-1 expression by hemin 48h before UUO attenuated fibrosis in rats by down-regulating inflammatory and pro-fibrotic genes, and decreasing pro-apoptotic pathways (caspase-3 activation), proteinuria and renal dysfunction [274,275]; ZnPP (an HO-1 inhibitor) prevented these protective effects. Confirming this protective role, HO-1 ^−/−^ mice with UUO exhibited increased fibrosis, tubular TGF-ß1 expression and inflammation, and had enhanced epithelial to mesenchymal transition with increased infiltration of macrophages [276], as well as increased expression of M2 markers and heavy-chain ferritin H [277]. HO-1 overexpression in transgenic mice was reported to significantly reduce renal interstitial fibrosis, inhibit the loss of peritubular capillaries, suppress activation and proliferation of myofibroblasts, limit the tubule-interstitial infiltration of macrophages and to regulate the secretion of inflammatory cytokines in UUO mice [278]. Taken together, these studies indicate a pleiotropic role for HO-1 in accelerated, obstruction-induced fibrosis, limiting macrophage infiltration while conferring upon them a regulatory phenotype, promoting microcirculation maintenance and inhibiting pro-fibrotic processes. It is, therefore, unsurprising that there is increasing interest in HO-1-targeted therapeutics to treat accelerated fibrosis.

Among the key maintenance factors important during CKD are the metabolic wastes and toxins, which the remaining nephrons are eventually unable to eliminate. Among them, indoxyl sulfate, a uremic toxin, has been shown to downregulate renal expression of Nrf2 through activation of NF-κB, thus down-regulating HO-1 and NQO1 and increasing production of ROS [307]. Indeed, ineffective HO-1 upregulation has been described in a mouse model of progressive post-ischemic inflammation [308], while a decreased Nrf2 expression (HO-1 transcription factor) was also reported in another mouse model of CKD [309,310] following upregulation of the Nrf2 repressor Keap1. This inefficient upregulation of HO-1 was also reported in patients, with significant suppression of plasmatic HO-1 upregulation after administration of low concentrations of SnPP in CKD patients versus healthy volunteers [311]. 

Interestingly, among patients with coronary artery disease, a greater number of GT dinucleotide repeats in the HMOX1 gene promoter, reducing its expression, has been associated with an increased risk of CKD [312]. Thus, the failure of anti-inflammatory enzymes such as HO-1 to be upregulated may provoke oxidative stress and inflammation, which in turn could contribute to this self-sustaining, injury-promoting state [308]. Despite impaired oxygen supply in CKD, HIF is down-regulated and has been reported as being involved in HO-1 downregulation: HIF induction in a mouse model of CKD restored HO-1 expression, together with VEGF and angiogenesis [313]. Finally, HO-1 in the proximal tubule may reduce albumin-stimulated production of cytokines such as MCP-1 [314]. Thus, repression of HO-1 expression by either Nrf2 or HIF seems to participate in CKD progression by unfavorably influencing angiogenesis and proteinuria.

To conclude, deficiencies in HO-1 upregulation during CKD may favor fibrosis, notably by decreasing angiogenesis and establishing and maintaining a pro-inflammatory state particularly with respect to the macrophage phenotype.

## 5. Modulation of the HO-1 Axis: Past, Current and Future Strategies 

Evidence of the health impact of HO-1 and its involvement in kidney diseases is accumulating and has motivated many researchers to investigate it as a therapeutic target. While this has led to some treatment suggestions, no drugs focusing on the HO system have yet been translated to clinic practice. The complexity of the disease processes studied, and sometimes conflicting results from animal models, contribute to difficulties in its clinical application. In this section, we will briefly describe recent and ongoing human clinical trials, before discussing some issues where HO-1 may be a promising therapeutic avenue. 

### 5.1. Past and Current Clinical Trials in Humans 

In the context of an acute exposure that exceeds the regulation systems, free heme is noxious. However, heme is also one of the main inducers of HO-1. Therefore, HO-1 upregulation by controlled administration of low doses of heme (or another HO-1 inducer) could result in a favorable biological response via the production of antioxidant and cytoprotective metabolites. The administration of heme as an HO-1 inducer has been considered as a therapeutic approach and, as described above, pre-treatment with low doses of heme in different animal models of kidney disease has been associated with a decrease in inflammation, oxidative stress and tissue damage [315,316,317]. In humans, both heme and heme arginate (a therapeutic agent for porphyria) induced HO-1 expression in healthy volunteers [318,319], while heme arginate improved experimental ischemia-reperfusion lesions in healthy individuals [320]. These data raised hopes for the therapeutic application of HO-1 induction using heme. However, hemin (the oxidized form of heme) treatment is not without adverse effects: after intravenous administration of hemin in healthy volunteers, thrombophlebitis occurred in 45% of cases (4/9 patients) [321]. Thrombotic complications have also been described in patients with intermittent acute porphyria treated with hemin infusion. Few studies of efficacy exist and their results are inconclusive. In a study of patients with metabolic syndrome, heme arginate did not improve endothelial function or insulin sensitivity, but significantly reduced the vasodilatory response to nitroglycerin [322]. These negative results contrast starkly with the preclinical data, possibly due to the short duration of treatment and limited induction of HO-1, as well as the interference of significantly elevated plasma heme levels [322]. Future studies should focus not only on accurately determining an effective dose and its timing, but also in fully describing the specific conditions under which the normalization of plasma heme levels is accomplished. Other molecules, such as statins [323,324] and 5-aminosalicylic acid [325], lead to positive regulation of HO-1. A clinical trial in a SCD population to evaluate the effect of atorvastatin on endothelial function and inflammatory and oxidative stress markers, including HO-1 activity, is currently under way (Table 2). 

The paucity of clinical trials targeting HO-1 in humans is noteworthy and Table 2 details the trials currently referenced on the ClinicalTrials.gov website. A search using the keywords “heme-oxygenase-1” AND “kidney” identified only 15 clinical trials, three of which had not had their status updated in more than two years, despite their completion date having passed, and were therefore not included in Table 2. 

### 5.2. Therapeutic Challenges

The apparent imbalance between the scale of research interest in HO-1 induction—both in vitro and in vivo—and the relatively few ongoing clinical trials is remarkable. Indeed, many gray areas remain regarding the consequences of HO-1 induction, hampering the transition from in vivo experiments to clinical trials. The long-term effects of HO-1 induction in tissues and cells are poorly understood and could have detrimental effects regarding, for example, tumorigenesis, efficacy of treatments for cancer or, more widely, susceptibility to infection.

Indeed, in contrast with all the protective effects described in kidney pathologies, HO-1 pro-survival properties favor cancer progression [326]. Indeed, enhanced HO-1 expression has been correlated with tumor growth, aggressiveness, metastatic and angiogenic potential, resistance to therapy, tumor escape and poor prognosis [327]. Furthermore, HO-1 inhibition showed benefits in a number of cancer models [328]. Even if HO-1’s role in cancer is probably dual (both protective and detrimental), chronic induction of HO-1 could play a role in tumor progression by aiding tumors to escape the host immune system, which remains to be properly studied.

HO-1 induction could also influence the efficacy of cancer treatments, which should be monitored in potential future studies. For example, HO-1 induction has been thoroughly studied to prevent cisplatin toxicity [8,23,94,202,203,204], which still frequently leads to acute kidney injury through glomerular, vascular and tubular damage [329]. Different molecules which increase HO-1 expression have been studied to prevent cisplatin-induced apoptosis and inflammation in the kidneys of mouse or rat models: microRNA-140-5p [206], JQ1 [207], farrerol [208], zinc oxide nanoparticles [209], human growth factor [210], bardoxolone methyl [205] and capsaicin (an active component of chili peppers) [211] have all been studied in this regard. Unfortunately, only a few of these articles reported upon the effect of these molecules on the proapoptotic, antitumoral action of cisplatin on tumoral cells [207], which is the major stumbling block for their use in offsetting iatrogenic renal damage in cancer patients.

Finally, immunoregulatory effects of HO-1 in circulating and resident monocytes and macrophages could influence sensitivity to infection. Indeed, all the pathologies described here, whether mediated by heme or not, are associated with an increased incidence of infections: in transplantation or autoimmune diseases, it is partly a result of immunosuppression; in chronic, heme-associated pathologies such as PNH (including before treatment by Eculizumab [330]), SCD, etc., recurrent infections are a hallmark of disease. HO-1 is thought to be essential for preventing tissue damage during infection; however, this role is dual—notably according to the pathogen type, as described for intracellular pathogens [331]—and its tolerogenic properties may reduce pathogen clearance [332]. Exploring to what extent the immunoregulatory functions of HO-1 could be immunosuppressive (perhaps only moderately so), in particular during long-term HO-1 induction, should be assessed in future studies to guarantee the safety of such applications. 

Thus, while in vitro and in vivo data show promising results in several kidney pathologies, more data on the potential bystander effects of long-term HO-1 induction are required to improve our understanding of how HO-1 could be targeted, and to enlarge the spectrum of clinical trials regarding HO-1 induction to other pathologies. The research interest over the past decade in other strategies, such as the administration of heme degradation products, reflects the need for researchers to bypass the duality of heme and HO-1 effects. Increasing levels of CO, through administration of CO-releasing molecules, has notably showed promising results in AKI and in improving transplantation success [333,334,335].

## 6. Conclusions

HO-1 has essential functions in preserving the kidney against injury, not only in heme-mediated pathologies, but also in a wide range of diseases which differentially affect glomeruli, vessels and tubules in an acute or chronic manner. While major benefits of its induction are expected, the wealth of HO-1 literature in vitro and in vivo contrasts with the small number of clinical trials under way to date. This suggests a need for data regarding the side effects of HO-1 induction, but also for more detailed studies aimed at developing therapeutic tools for specific induction of HO-1 in different structures/cell types.

## Figures and Tables

**Figure 1 ijms-22-02009-f001:**
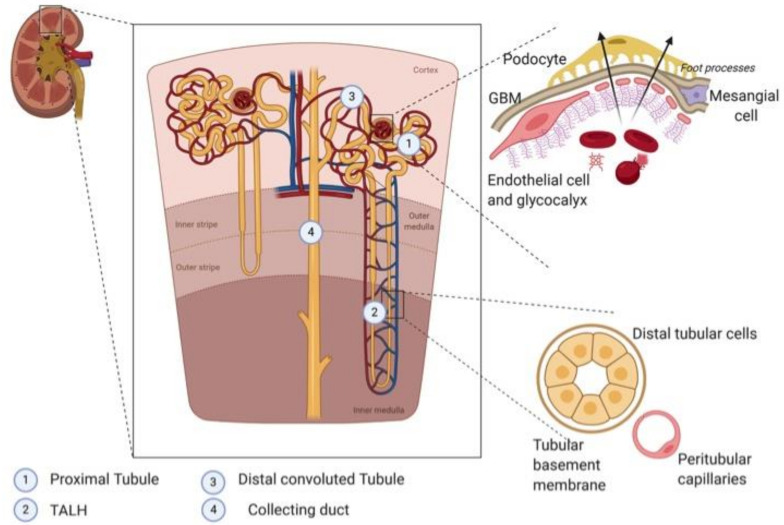
Schematic view of kidney structures describing cortical and medullar compartments, components of glomerular basement membrane and tubular segments and structures. GBM: Glomerular basement membrane, TALH: Thick ascending loop of Henle (Created with BioRender.com).

**Figure 2 ijms-22-02009-f002:**
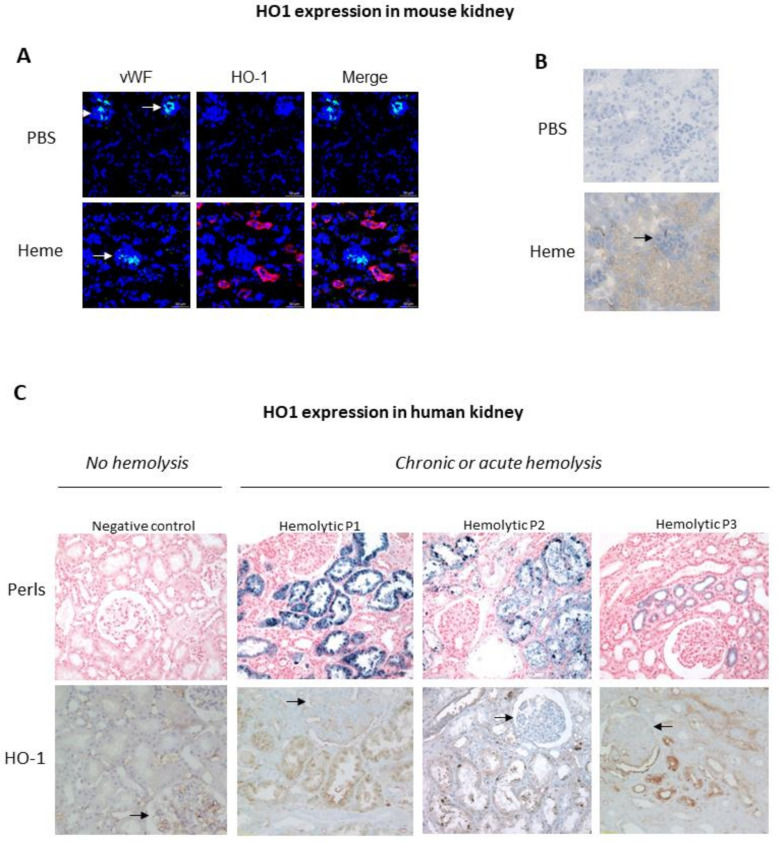
Renal expression of HO1 (adapted from [27]). HO-1 staining in mouse kidneys in IF (**A**) and IHC (**B**). A- HO-1 (red) staining, vWF (green) staining, and colocalization on frozen kidney (x15) sections of mice, injected with PBS (upper panel) or heme as HO1 inducer (lower panel), studied by IF (**A**); B- HO-1 staining appears in brown on frozen kidneys sections of mice treated with PBS or heme. (**C**) HO-1 expression in human kidneys: hemosiderin deposits in human kidney biopsies evaluated by Perls’ coloration and HO-1 (brown) staining were performed by IHC. A normal protocol, kidney allograft biopsy performed 3 months after transplant was used as a negative control. Patients 1 and 2 suffered chronic hemolysis associated with prosthetic heart valves, and patient 3 carried the C3 mutation complicated with atypical hemolytic uremic syndrome (arrows indicate glomeruli).

**Figure 3 ijms-22-02009-f003:**
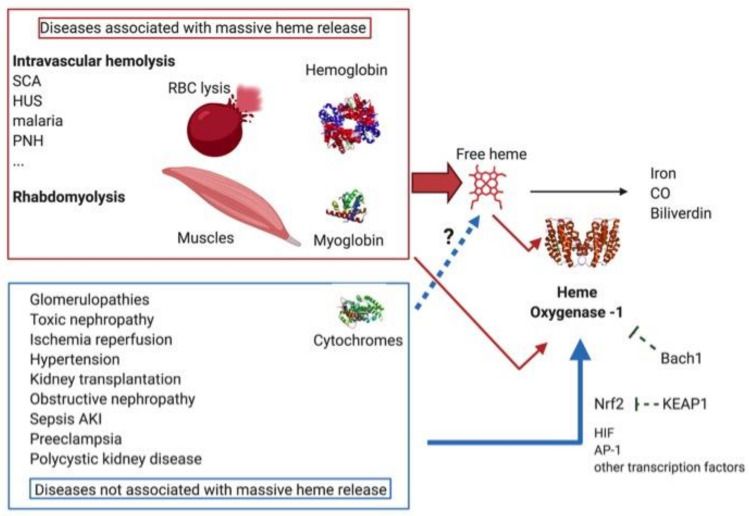
Schematic description of HO-1 regulation in renal pathologies. Plain arrows: proved link, Dotted arrows: suspected link, AKI: acute kidney injury, SCA: sickle cell anemia, HUS: hemolytic uremic syndrom, PNH: paroxysmal nocturnal haemoglobinuria (Created with BioRender.com).

**Table 1 ijms-22-02009-t001:** Main models and functions of HO-1 in kidney pathologies.

Pathology			Number Pub.	Models	HO-1 Main Functions in the Pathology	References
**Heme dependent**	**Hemolysis**	Sickle cell disease	14	Mice (Townes SS) EC + MPs from SCA patients βS/βS-antilles mice + SnPP	Preserves local kidney hemodynamics Decreases exposure to oxidative stress	[123,124,125,126,127,128,129,130,132,137]
		Hemolytic Uremic Anemia	4	Microvascular glomerular cells + heme	Weakly expressed in glomerular EC increasing its susceptibility	[26,64,141,142]
		Malaria	5	HO-1 −/− mice + *P. berghei* ANKA Mice + *P. berghei* ANKA +/− FePP or CoPP or ZnPP HO-1 −/− mice + *P. chabaudi chabaudi*	Decreases proximal tubular necrosis	[143,144,145]
		Paroxysmal nocturne hemoglobinuria	4	ns	unknown	[153,154]
	Rhabdomyolysis		24	Mice +/− Hb or heme + glycerol HO-1 −/− mice + glycerol	Improves renal microcirculation Reduces oxidative stress Regulates immune response	[33,111,162,163,164,165,166,167]
Not heme dependent	Glomerular nephropathies	Lupus	13	MRL/lpr mice + hemin MRL/lpr bach1−/− mice Nrf2−/− mice Pristane-induced LN mice + Nrf2 activators Patients’ mph, Mo, PNN	Reduces proteinuria, Decreases proliferation and deposition in glomeruli Is weakly expressed in renal and circulating monocytes/mph	[177,178,179,180,181]
		Anti-GBM	6	Anti-rat nephrotoxic globulin + hemin Rat +/− hemin + anti-GBM	Reduces proteinuria Induces switch to a M2 macrophage phenotype	[182,183,184]
		Membranous nephropathy	4	Albumin induced membranous nephropathy + CoPP	Reduces proteinuria Improves glomerular lesions Decreases immune-complex deposition, oxidative stress,	[185,186]
		Diabetes	136	STZ- induced DM mice/rat +/− HO-1 siRNA +/− hemin +/− ZnPP or CoPP or SnMP Genetic diabetic db/db mice Diabetic fat rat	Decreases albuminuria, Limit podocytes apoptosis, Decreases mesangial proliferation, Modulate renal hemodynamics, Preserves insulino-sensitivity Influences glomerulo-tubular crosstalk in a heme-dependant manner.	[72,187,188,189,190,191,192,193,194,195,196,197,198,199,200,201]
	Toxic nephropathy	Cisplatin toxicity	86	Rat +/− SnPP + cisplatin Targeted HO-1 deletion in PT of mice (CreLox) + cisplatin HO−/− mice + cisplatin HO+/+ mice +/− hemin + cisplatin	Improves renal function Decreases tubular epithelial cells – notably proximal - necrosis Increases autophagy	[23,94,202,203,204,205,206,207,208,209,210,211,212]
		Gentamycin toxicity	7	Rat +/− SnPP + gentamycin HO-1 overexpressing or KD NRK-52E cells Mice overexpressing HO-1 in kidney (adenoviral construct)	Decreases apoptosis in tubular cells	[202,213,214]
		Ciclosporin toxicity	35	LLC-K1 + CsA +/− SnPP NRK-52E + CsA Nrf2 −/− mice + CsA	Modulates EMT Decreases PT cells apoptosis and oxidative stress	[215,216,217,218,219,220]
		Rapamycin toxicity	13	Rat transplantation+Rapamycin Mice with pedicle ligation + rapamycin	Preserves kidney dysfunction	[221,222,223]
		Radiocontrast product	3	Rat +/− SnPP or CoPP + sodium iothalamate	Preserves kidney function+ decreases apoptosis	[224]
		Mercuric chloride toxicity	7	Mice +/− hemin+/− HgCl2	Improves renal function and Decreases distal tubule necrosis	[225,226,227]
	Ischemia reperfusion		183	Unilateral/bilateral renal arteries clamping in - HO−/− mice - HO+/+ mice preventively treated by SnPP/hemin - mice overexpressing HO-1in BMDM (adenoviral construct) - myeloid-restricted deletion of HO-1 mice (Cre-Lox) HO-1 overexpressing LLC-PK1 cells	Improves renal function -in tubular cells, * modulating local hemodynamic through NO synthesis regulation * conferring resistance to apoptosis - in myeloid cells * preserve kidneys from their pro inflammatory effects	[91,103,205,221,228,229,230,231,232,233,234,235,236,237,238,239,240,241,242,243,244,245,246,247,248,249,250,251,252,253,254,255]
	Kidney transplantation		215	Hemin or CoPP in rat donor transplant CoPP or ZnPP in rat or mouse transplant receivers	Prevents acute rejection Prevents IS toxicity In donor, preserves kidney graft function and prevents post reperfusion apoptosis	[39,221,251,252,253,254,256,257,258,259,260]
	Hypertension		146	Unilateral kidney artery clamping in HO-1 −/− mice Spontaneously hypertensive rats +/− heme +/− ZnPP Ang II induced hypertension in HO-1 −/− mice HO-1-specific overexpression in TALH	Not essential to preserve arterial blood pressure in physiological conditions Reduce blood pressure under pathological conditions, (increase of Angiotensin II) In TALH: maintains blood flow and natriuresis	[25,261,262,263,264,265,266,267,268,269,270,271]
	Obstructive nephropathy		43	Unilateral or bilateral obstruction * +/− preventive heme +/− ZnPP * in HO-1 −/− mice * in transgenic mice overexpressing HO-1	Limits macrophages infiltrate Confers mph a regulatory phenotype, Promotes microcirculation maintenance Inhibits profibrotic processes	[272,273,274,275,276,277,278]
	Sepsis-AKI		46	HO-1 −/− mice + LPS HO-1−/− mice + cecal ligation Myeloid-restricted deletion of HO-1 mice (Cre-Lox) + *Enteroccocus faecalis*/*Escherichia coli*	Preserves glomerular filtration rate Preserves renal blood flow, Decreases renal cytokine expression Decreases tubular necrosis Degrades heme In macrophages, contributes to pathogen clearance and decreased AKI	[105,279,280,281]
	Preeclampsia and kidney		4	HUVEC +/− overexpressing HO-1 (adenoviral construct) + PAR2 activating peptides HUVEC + VEGF-E Reduced uterine perfusion pressure rats +/− CoPP	Promotes proper placental vascular development Decreases endothelial PAR-2 activation Reduces Edothelin-1 production (in GEC) Attenuates hypertension	[282,283,284,285]
	Polycystic kidney disease		6	Cys1cpk/cpk mice + CoPP or SnPP	Decreases kidney weight Decreases kidney injury markers Decreases cystogenesis	[286,287,288]

Number of articles referenced in Pubmed.ncbi.nlm.nih.gov, identified by the key words "heme-oxygenase-1" AND ""Kidney"", and the name of the pathology. AKI: acute kidney injury; Ang II: Angiotensin II; CoPP: cobalt protoporphyrin; EC: endothelial cells; EMT: epithelial–mesenchymal transition; FePP: iron protoporphyrin; GBM: glomerular basement membrane; GEC: glomerular endothelial cells; HUVEC: human umbilical vein EC; IS: immunosuppressors; Pub.: publication; LPS: lipopolysaccharide; Mo: monocytes; mph: macrophages; SnMP: tin mesoporphyrin; SnPP: tin protoporhyrin; ZnPP: zinc protoporphyrin. Asterisks (*) and hyphens (-) indicat detailed explanations of functions or models, the right above ones.

**Table 2 ijms-22-02009-t002:** Clinical trials for HO-1 and kidney.

Study Topic	Population	Study Design	Intervention/Treatment	Outcome Measures	ClinicalTrials.gov Identifier	Sponsor
SnPP in Healthy Volunteers and Stage 3–4 CKD patients	Healthy volunteers Stage 3–4 CKD	Interventional no-random.	A single dose of 9, 27 or 90 mg of SnPP (followed for 28 days)	Measuring Haptoglobin, Ferritin, Bilirubin, Hemopexin, IL-10, and **HO-1**, the P21 biomarker systemic levels	NCT04072861	Renibus Therapeutics, Inc.
RBT-1 in Healthy Volunteers and Stage 3–4 CKD patients	Healthy volunteers Stage 3–4 CKD	Interventional no-random.	A single dose of 240 mg Iron Sucrose and 9, 27 or 90 mg of SnPP (followed for 28 days)	Change in biomarkers of cytoprotective activity: Haptoglobin, Ferritin, Bilirubin, Hemopexin, IL-10, **HO-1**, the P21 biomarker systemic levels	NCT03893799	Renibus Therapeutics, Inc.
Effects of Resveratrol in CKD Patients	Stage 3 - 4 CKD	Interventional random. CO	500mg of trans-resveratrol per day for 4 weeks vs. placebo	Antioxidants and anti-inflammatory biomarkers: **Nrf2**, GPx, **HO-1**; NFkB, IL-6, TNF-alfa	NCT02433925	Universidade Federal Fluminense
Propolis in CKD Patients	CKD under conservative treatment	Interventional random. CO	20 drops of standardized propolis daily for 2 months vs. non-intervention.	Change in cytokines plasma levels, **Nrf2** and NF-kB, antioxidant enzymes (NQO1, **HO-1**), NLRP3 receptor, PPAR-γ. Change the profile of the intestinal microbiota	NCT04411758	Universidade Federal Fluminense
Chocolate in CKD Patients	CKD in hemodialysis	Interventional no-random.	40g dark chocolate per day for 2 months vs. no intervention	Change in antioxidants and anti-inflammatory biomarkers: **Nrf2**, GPx, **HO-1**, NFkB, IL6, TNFa. Uremic toxins.	NCT04600258	Universidade Federal Fluminense
Effects of Curcumin Supplementation in HD patients	CKD in hemodialysis	Interventional random. CO,	3 capsules per day with 500mg of curcumin and 5mg of piperine for 4 weeks vs. placebo.	Change in antioxidants and anti-inflammatory biomarkers: **Nrf2**, glutathioneperoxidase (GPx), **HO-1**, NFkB, IL6, TNFa	NCT03475017	Universidade Federal Fluminense
Effects of Curcumin Supplementation in PD Patients	CKD in PD	Interventional random. CO	3 capsules with 500mg of curcumin and piperine per day, for 12 weeks vs. placebo	Change in antioxidants and anti-inflammatory biomarkers: **Nrf2**, GPx, **HO-1**, NFkB, IL6, TNFa, PCR, IL-18, Inflammasome.	NCT04413266	Universidade Federal Fluminense
Tacrolimus After rATG and Infliximab Induction IS	KTR	Interventional no-random.	rATG induction (2 × 1.5 mg/kg) in 1st KTR given perioperatively and on first postoperative day, then Infliximab mAb at day 2.	Composite endpoint of efficacy failure and renal function of the induction regimen at 12 months post transplantation. **HO-1 polymorphisms**	NCT04114188	Charite University, Berlin, Germany
Induction of HO-1 to Reduce Ischemia Reperfusion Injury in KTR	KTR	Interventional Random.	Heme arginate (3mg/kg) or placebo prior to transplant and repeat again on day 2 post-transplant	Measure the level of **HO-1** protein in isolated macrophages/monocytes in a blood sample taken at 24 h after drug infusion	NCT01430156	University of Edinburgh
Characterization of the Nrf2 Response in Patients With ADPKD	ADPKD	Observational Prospective	No intervention	8-oxodeoxyguanosine, F2-isoprostanes, **HO-1**, SOD, catalase, GPx, NAD(P)H dehydrogenase, glutathione, **Nrf2**.Total kidney volume determined by MRI.	NCT04344769	Mayo Clinic
Effect of Atorvastatin on Endothelial Dysfunction and Albuminuria in SCD	SCD	Interventional random. CO	Atorvastatin 40 mg once daily for 6 weeks vs. placebo.	Change from Baseline to Week 6 in endothelial function, in plasma markers of endothelial activation, in **HO** activity.	NCT01732718	University of North Carolina, Chapel Hill

Clinical trials referenced in ClinicalTrial.gov, identified by the key words “heme-oxygenase-1” AND “Kidney” (OR “renal”). ADPKD: autosomal dominant polycystic kidney disease; CKD: chronic kidney disease; CO: crossover; GPx: glutathioneperoxidase; GSR: glutathione reductase; KTR: kidney transplant recipient; HD: hemodialysis; PD: peritoneal dialysis; PPAR-γ: peroxisome proliferator-activated receptor-γ; Random.: randomized; SnPP: stannous protoporphyrin; SOD: superoxide dismutase. Bold letters present emphasis and highligt.

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
