# Peer review of "Heme Oxygenase 1: A Defensive Mediator in Kidney Diseases"

_ijms, 2021, doi:10.3390/ijms22042009_

Round 1
Reviewer 1 Report
The paper by Grunenwald et al. is an exhaustive review addressing the regulation and role of the HO system in renal diseases. It is meticulously prepared and comprehensively describes the knowledge arising from hitherto studies. However, it may be considered as too extensive for some Readers, who look for a summary overview of HO-1 role. It would be worth adding some summary sentences and pictures to outline the most important findings/pathways.
Some parts need clarification and more consistency. Please review at least those sentences for clarity and brevity:
page 2: "The particular susceptibility of the kidney to dysfunctions in the HO-
1 system is in part related to its physiological functions: it is very susceptible to hypoxia [12], one of major drivers of the expression of HO-1, and in addition the kidney is pivotal for the filtration and reabsorption functions of the organism".
"Its expression is dependent on the HO-1 gene (HMOX1)." (page 3)
"This lower capacity of glomeruli to express HO-1, one of the key cytoprotective mechanisms, could explain their particular susceptibility in certain pathologies [27]." (page 4)
"In their model, rapid induction of HO-1 messenger RNA and protein was observed within the kidney by a single prior injection of hemoglobin, which prevented progression to renal failure [34]." (page 5)
"These results echo the susceptibility of elderly humans to
nephrotoxic agents." (page 6)
"Sickle cell anemia (SCA) is characterized by a mutated Hb gene (HbS) responsible for the polymerization and deformation of erythrocytes, thereby reducing their lifespan [101]." (page 13)
"On the other hand, some cytoprotective effects against glycerol-induced RIAKI may be independent of HO-1 expression, for example with treatment by N-acetylcystein (a scavenger of free radicals) [169]". (page 16)
"Preventive HO-1 induction, together with induction of other anti-inflammatory and cytoprotective stress proteins such as ferritin, haptoglobin, hemopexin, alpha 1 antitryp-sin and IL-10, has been proposed as a preconditioning treatment, notably in a glycerol-induced RIAKI model. For example, infusion of nitrated myoglobin or SnPP played a syn-ergistic role in HO-1 induction, thus" (page 16)
"HO-1 inducers have generated significant interest in animal models of I/R and many of them have shown protective effects. Among them, hemin exhibited conflicting effects, probably owing to the timing and concentration of administration, namely protective at low doses and before Ischemic injury but deleterious in other cases [180,184]." (page 17)
"Indeed, inefficient HO-1 upregulation has been described in a mouse model of progressive post-ischemic inflammation [266], as well as impaired expression of nrf2 (HO-1 transcription factor) in another mouse model of CKD [267,268]" (page 21)
"Future studies should pay attention to the delicate balance between adequate dosing and timely normalization of plasma heme levels." (page 22)
"Indeed, in contrast with all the protective effects described in kidney pathology, HO-9 1 exerts an important role in cancer progression, due to its pro-survival properties [284]. 10 HO-1 expression has been correlated with tumor growth, aggressiveness, metastatic- and 11 angiogenic potential, resistance to therapy, tumor escape and poor prognosis [285], and 12 its inhibition showed benefits in a number of cancer models [286]." (page 26)
Please revise the spelling using the English names (ex. page 11 - diabete, rapamycine etc)
Please use the abbreviations consistently - nrf2/Nrf2
Author Response
Response to reviewers regarding ijms-109719
We sincerely thank the Reviewers for their comments, and we are grateful for the positive evaluation of our work.
In response to the reviewer 1 concerns, we have added summary sentences at the end of paragraphs where it was missing and designed a new overview figure regarding the most important findings and pathways. We revised the text as proposed regarding clarity and brevity.
All changes that we made are visible in revised manuscript
Reviewer 2 Report
Paper solid and well written, interesting content. To me, it can be published in the present form.
Author Response
Response to reviewers regarding ijms-109719
We sincerely thank the Reviewers for their comments, and we are grateful for the positive evaluation of our work.
All changes that we made are visible in revised manuscript
Round 2
Reviewer 1 Report
The manuscript has improved and may be considered for publication.